

# Improved neural network Monte Carlo simulation

I-Kai Chen[1], Matthew D. Klimek[1,2]* and Maxim Perelstein[1]

**1** Laboratory for Elementary Particle Physics, Cornell University, Ithaca, NY, USA
**2** Department of Physics, Korea University, Seoul, Republic of Korea

* klimek@cornell.edu

## Abstract

The algorithm for Monte Carlo simulation of parton-level events based on an Artificial Neural Network (ANN) proposed in Ref. [1] is used to perform a simulation of $H \rightarrow 4\ell$ decay. Improvements in the training algorithm have been implemented to avoid numerical instabilities. The integrated decay width evaluated by the ANN is within 0.7% of the true value and unweighting efficiency of 26% is reached. While the ANN is not automatically bijective between input and output spaces, which can lead to issues with simulation quality, we argue that the training procedure naturally prefers bijective maps, and demonstrate that the trained ANN is bijective to a very good approximation.

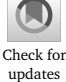

# 1 Introduction

Monte Carlo (MC) simulations of high energy particle collisions play a central role in particle physics. Interpretation of large data sets that will be collected in the upcoming runs of the Large Hadron Collider (LHC) will demand MC samples of unprecedented size and accuracy. Improving the efficiency of numerical algorithms used in MC simulations is an important and timely task, see *e.g.* [2]. In this paper, we focus on the most basic task of an MC simulation: generating a set of points distributed according to a known probability distribution function (pdf) on a target phase space. The application we have in mind is generation of parton-level events, and the pdf is the fully differential cross section (or decay width) which we assume to be known either analytically or numerically prior to the simulation.[1] Today's general-purpose MC tools, such as MadGraph [4], Herwig [5], and Sherpa [6], rely on improved versions of the original VEGAS algorithm [7,8] to perform this basic task. Recently, it has been suggested that machine learning algorithms, in particular those based on artificial neural networks (ANN), can offer significant advantages [1,9–17]. In Ref. [1], two of us (MDK and MP) have demonstrated that in simple applications, including processes with resonances and infrared/collinear singularities with up to three particles in the final state, an ANN-based MC algorithm achieved significantly higher unweighting efficiency (up to a factor of 10 improvement) compared to existing general-purpose tools. Another important advantage of the ANN-based approach, also demonstrated in [1], is that features such as resonances do not need to be aligned with any of the chosen coordinate axes on phase space to be simulated efficiently, in contrast to grid-based algorithms such as VEGAS. In cases containing multiple features that cannot be simultaneously aligned with the grid, such algorithms require the use of supplementary techniques such as multi-channeling [18–20]. The ANN-based algorithm, however, can flexibly adapt to multiple features of various shapes simultaneously.

The goal of this paper is to build upon Ref. [1] to further develop and demonstrate the ANN-based approach to MC simulation. The previous work dealt with simulations of toy-model processes. Here, we consider a fully realistic example of high relevance at the LHC, namely the Higgs decay to four charged leptons. Experimentally, this process has been crucial in confirming the quantum numbers of the Higgs, and provided a sensitive measurement of the Higgs coupling to the $Z$ boson [21–24]. We show that the ANN-based MC algorithm can provide an efficient and accurate simulation of this decay, including a faithful representation of its non-trivial resonance structure.

Before presenting the results, let us comment on the relation of our work to some of the recent papers on ANN-based MC algorithms. The literature can be divided into two classes. One approach [10–12, 17] is to start with an existing MC sample, generated for example by a general-purpose tool, train a neural network (typically a Generative Adversarial Network or GAN) to reproduce this sample, and then use the GAN to generate a larger sample at a lower computational cost. In this case, the GAN is used to essentially inter/extrapolate the distribution generated by another tool. For a discussion of potential limitations of this approach, see [25, 26]. Another approach [1,9,13–15] is to perform a self-contained first-principles simulation, by taking the invariant matrix element as an input and populating the phase space according to $|\mathcal{M}|^2$. In this paper, we follow the self-contained route and generate MC samples based on explicitly known matrix elements with no need for a prior independent MC simulation.

Refs. [14, 15] pursue an approach similar to ours, but use normalizing flows, rather than fully-connected ANNs, to perform the simulation. An important advantage of this algorithm is that the mapping used in the simulation is automatically bijective. In our setup, bijectivity is

---

[1]Our approach to parton-level simulation would work equally well if the target pdf were instead chosen to obtain a sample of weighted events optimized for a specific analysis, as suggested in Ref. [3].

not guaranteed. Lack of bijectivity can lead to issues with phase space coverage, accuracy of unweighting procedure, *etc.* To address this issue, we have developed techniques to test the trained ANN for bijectivity *a posteriori*. Using these techniques, we confirmed that the trained ANN in the example considered in this paper is indeed bijective to a good approximation.

While both of the normalizing flow studies [14,15] find marked improvement in efficiency over VEGAS for processes with three particles in the final state, the reported performance drops to a level comparable with VEGAS upon adding a fourth particle. These studies represent some of the earliest attempts in this direction, and further study is certainly warranted and likely to result in continued improvement. However, in this work we will consider a process with four final state particles, and show that substantial improvement over VEGAS is obtained straightforwardly. Although further study would be needed to rigorously characterize the performance of these various techniques, we will briefly make note of one basic feature of the normalizing flow approach which they have in common with VEGAS. In both cases, the map from the input space onto phase space is composed of a finite number of discrete intervals with a fixed order interpolation used in each interval. In VEGAS this is all there is. In the normalizing flow approach, there are several layers of such maps, with each acting on a different subset of coordinates, and with the parameters of each map controlled by an ANN trained to minimize the error. This approach is clearly much more flexible than VEGAS. Nevertheless, the ability to represent the target distribution must ultimately be limited by the finite number of intervals that are used, and this may be exacerbated as one goes to larger numbers of phase space dimensions. In contrast, our method is fully continuous (see, for example, the discussion in [1]), and in that sense may not suffer from the same kind of limitations as the normalizing flows.

The rest of the paper is organized as follows. In Sec. 2, we review the basic structure of ANN-based MC algorithm introduced in Ref. [1], and discuss improvements in the training procedure necessary to handle issues that arise for a 4-body phase space. Sec. 3 describes a systematic way of parametrizing a 4-body phase space as a 5-dimensional hypercube, the natural choice for ANN output space. Sec. 4 contains the results of ANN-based simulation of the on-shell Higgs decay into four charged leptons. In Sec. 5, we discuss issues related to the bijectivity of the map represented by the ANN. Finally, we conclude in Sec. 6.

## 2 Neural Network Setup and Training

Our Monte Carlo algorithm is based on an artificial neural network, which can be thought of as a highly non-linear, adjustable map from an input space $I$ to the target space $T$; see Fig. 1. In our application, the target space is identified with phase space. Both input and target spaces are unit hypercubes with dimensionality equal to the number of relevant phase space dimensions. The dimensionality of input and target spaces matches the number of nodes in the input and output layers of the ANN. The main idea is to train the ANN so that it maps a uniform sample of the input space points $\{\mathbf{x}\}$ into a set of phase space points $\{\mathbf{y_w}(\mathbf{x})\}$ distributed according to the known target pdf $f(\mathbf{y})$. The target pdf is the differential cross section or decay width of the process at hand, *i.e.* a product of the invariant matrix element-squared $|\mathcal{M}|^2$, and phase space volume factor in the coordinate system used to parametrize $T$. The phase space density induced by the ANN is given by

$$p_y(\mathbf{y}) \equiv p_y(\mathbf{y_w}(\mathbf{x})) = \left| \frac{\partial y_i}{\partial x_j} \right|^{-1} . \tag{1}$$

Training the ANN consists of adjusting its parameters[2] $\mathbf{w}$ such that $p_y(\mathbf{y}) \propto f(\mathbf{y})$. To achieve this, the loss function is defined to be the Kullbeck-Leibler (KL) divergence between the two

---

[2]The adjustable parameters for a fully-connected ANN include weights and biases.

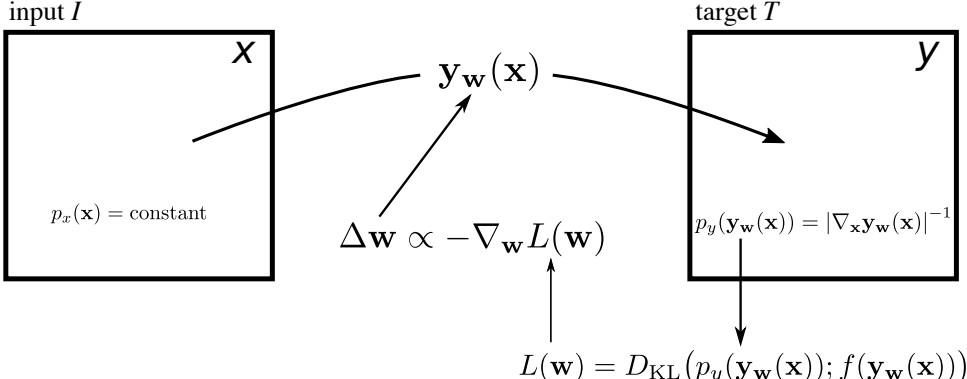

$$L(\mathbf{w}) = D_{\text{KL}}\big(p_y(\mathbf{y_w}(\mathbf{x})); f(\mathbf{y_w}(\mathbf{x}))\big)$$

Figure 1: The ANN is a map $\mathbf{y_w}(\mathbf{x})$ between a uniformly sampled input space and a target space on which it induces a non-trivial pdf. During training, the parameters $\mathbf{w}$ of the ANN are adjusted to minimize loss function $L(\mathbf{w})$, the KL divergence between the induced and target pdfs. From Ref. [1].

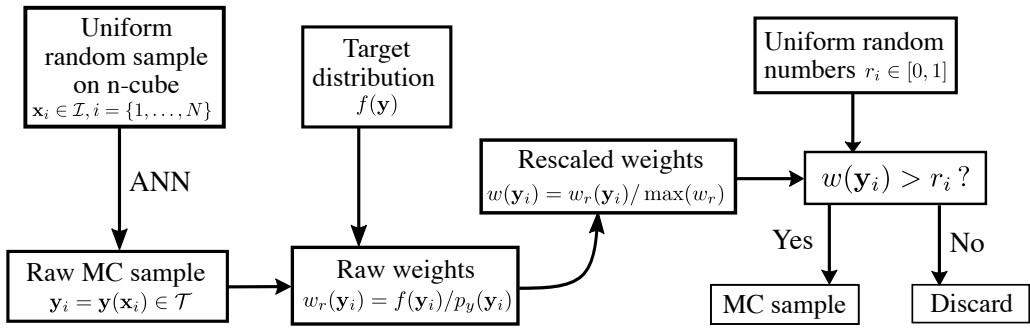

Figure 2: Diagram representing an ANN-based Monte Carlo generator. From Ref. [1].

distributions:

$$L_{\text{KL}}(\mathbf{w}) = D_{\text{KL}}[p_y(\mathbf{y}); f(\mathbf{y})] \equiv \int p_y(\mathbf{y}) \log \frac{p_y(\mathbf{y})}{f(\mathbf{y})} \, d\mathbf{y}. \tag{2}$$

At each step (or "epoch") in the training process, the gradient of the KL divergence with respect to $\mathbf{w}$ is evaluated numerically using a batch of random input space points. The gradient is then used by the `Adam` algorithm [27] (a variant of the gradient-descent method) to update the parameters. Repeating this process iteratively yields a numerical solution to the minimization problem for the loss function, which corresponds to the best possible approximation to $p_y(\mathbf{y}) \propto f(\mathbf{y})$.

After training is completed, the ANN parameters $\mathbf{w}$ are frozen, and the ANN is used as the engine for an MC generator described in Fig. 2. The sample of "raw" phase-space points produced by the ANN is further improved by the unweighting procedure, which discards some of the generated points to achieve better representation of the target pdf. Specifically, for each point in a large sample of size $N$ generated by the ANN, we compute a raw weight $w_r(\mathbf{y}) = f(\mathbf{y})/p_y(\mathbf{y})$. If the probability distribution $p_y$ induced by the ANN overestimates the target distribution $f$ in the neighborhood of some point in the sample, we will have $w_r < 1$. In order to rectify this, only a fraction of points proportional to $w_r$ in that region should be retained. On the other hand, if in some region the ANN underestimates the tar-

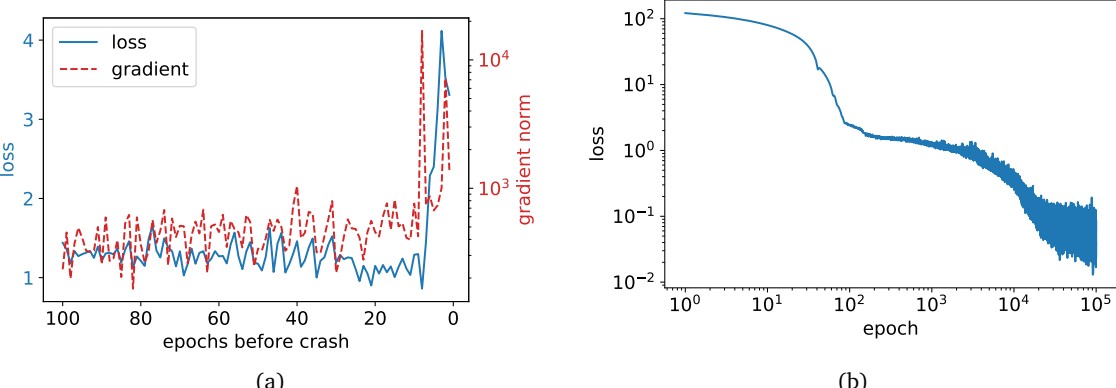

Figure 3: (a) During training, sampling of a region with $f(\mathbf{y}) \approx 0$ leads to a sharp jump in the gradient norm $|\boldsymbol{\nabla} L_{\mathrm{KL}}(\mathbf{w})|$, a subsequent rise in the loss function, and a division-by-zero attempt, causing the code to crash. (b) After gradient clipping is introduced, the numerical instability is eliminated and training proceeds smoothly until the minimum of the loss function is found.

get distribution so that $w_r > 1$, one must keep all points there and scale down the retained fraction in other regions in order to maintain the correct shape of the distribution. Therefore every generated point should be retained with a probability equal to the rescaled weight $w(\mathbf{y}) = w_r(\mathbf{y})/\max(w_r)$, where $\max(w_r)$ is the maximum raw weight that was observed in the sample. This unweighting process corrects the output of the ANN at the expense of inefficiency due to computational resources wasted in creating the discarded points. This is quantified by the unweighting efficiency, the fraction of points in the raw sample that are retained after unweighting, which is given by the average value of the rescaled weight over the whole sample

$$\mathcal{E} = \frac{1}{N} \sum_{\{\mathbf{y}_i\}} w(\mathbf{y}_i). \tag{3}$$

If the ANN's distribution exactly matches the target then all raw weights will have the same value giving $\mathcal{E} = 1$. We therefore use the unweighting efficiency as a measure of how well the trained ANN reproduces the desired phase space distribution.

The simulations presented in this paper use a fully-connected ANN architecture, with 6 hidden layers of 64 nodes each, implemented in MXNet [28]. We use the exponential linear unit (ELU) as the activation function and the soft clipping function ($SC_p$) introduced in [1] as the output function. The input and output layers have 5 nodes each, matching the number of non-trivial dimensions in the 4-body phase space of the $h \rightarrow 4\ell$ decay. The ANN is trained with batches of 1,000 points each, drawn uniformly from the input space. Our earlier study [1] used a batch size of 100 for 2 non-trivial dimensions in the 3-body phase space. We found that due to the increase in dimensionality in the 4-body case a larger batch size was necessary to adequately sample phase space in each training epoch. This allowed the training to obtain a lower minimum in the loss function. A batch size of 10,000 was also tested. However, this reduced the training speed in the initial phase of training (the first downward slope in Fig. 3 (b)). The size of the batches was therefore determined by a compromise between the need to comprehensively sample the phase space at each epoch and the available computational resources.

An additional complication may arise if the target pdf vanishes along some sub-manifold of phase space, leading to a numerical instability in the evaluation of the loss function. (In our example, the target pdf vanishes along one of the phase space boundaries due to a coordinate

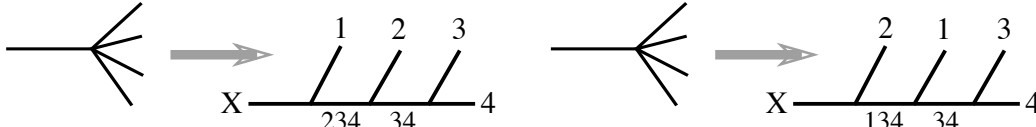

Figure 4: Two decompositions of a 4-particle decay into 2-particle decay chains for the definition of various kinematic invariants.

singularity; for details, see Sec. 3. Such singularities did not appear in the two- and three-body final states studied in Ref. [1], and hence the instability was not encountered in that work.) During training, the ANN may be in a state such that it induces non-zero probability density in a region where $f(\mathbf{y})$ is very small. When that region is sampled during training, the loss function and its gradient with respect to the weights at that point will be very large, according to Eq. (2). The training algorithm will then make a very large change in the ANN weights, which can take the ANN far away from the desired state. Such sudden jumps often cause the ANN to sample points so close to the phase-space boundary that $f(\mathbf{y}) = 0$ within machine precision, resulting in a division by zero error. We illustrate this behavior in Fig. 3(a), which shows the gradient norm and the loss function for the last 100 epochs before this error occurs. The gradient norm increased more than one order of magnitude at about 10 epochs before the error, and the loss function increased subsequently.

To avoid this instability, we modified the training algorithm by imposing an upper limit on the norm of the gradient $|\nabla L_{\mathrm{KL}}(\mathbf{w})|$ used in each training step. Any gradient with a norm greater than the limit, which we set at $10^4$, is rescaled down in order to avoid sudden jumps in the ANN parameter space, while its direction is preserved. With this modification, the training procedure is stable and good agreement of the trained ANN with the target pdf can be achieved. The loss function of a typical training run of the ANN after imposing gradient clipping is shown in Fig. 3(b). We find that the training converges after $10^4$–$10^5$ epochs. The loss function remains stable throughout the training process.

## 3   Phase Space Parametrization

The ANN maps a uniform distribution on a unit hypercube onto a non-trivial distribution on another unit hypercube which represents the phase space for the process under consideration. As such, an important feature of our setup is the way in which phase space is coordinatized in terms of the natural coordinates of the hypercube. Since we would ultimately intend for this ANN algorithm to form part of a general-purpose MC generator, this should be done in a way which is agnostic to the process and easily generalizable to any number of final state particles. Such a prescription was given in [1]. In this section we will review this prescription and give its detailed form as applied to the 4-particle case.

Given $N$ particles in the final state of some decay or scattering process, we form $N-2$ subsets: $\{\{1,\ldots,N-1\},\ldots,\{1,2\}\}$. The invariant masses of these subsets will serve as $N-2$ of the phase space coordinates. Then $2N-5$ relative angles can be chosen between the momenta of various final state particles. Lastly, an overall rotation can be specified by choosing three Euler angles. In total, this prescription provides $(N-2)+(2N-5)+3 = 3N-4$ coordinates needed to parameterize $N$-particle phase space in three spatial dimensions with the constraint of energy-momentum conservation. Finally, the coordinates can be rescaled so that each ranges from 0 to 1, projecting the phase space onto a unit hypercube.

For a 4-particle final state, our prescription dictates that we choose the invariant masses of one triplet, which we can take to be $\{2, 3, 4\}$, and one pair, for example $\{3, 4\}$. This corresponds to the decomposition shown in the left panel of Fig. 4. (Note that the internal lines in this figure do not necessarily correspond to physical resonances.) The phase space weight then contains the product of several 2-body factors in terms of the intermediate invariant masses corresponding to the chosen decomposition:

$$d\Pi_4 = m_X^{-1}\lambda(m_X; m_1, m_{234})\lambda(m_{234}; m_2, m_{34})\lambda(m_{34}; m_3, m_4)\, dm_{234}\, dm_{34}\, d\Omega^{(X)}\, d\Omega^{(234)}\, d\Omega^{(23)}\,, \tag{4}$$

where $d\Omega^{(S)}$ represents the uniform measure on the sphere in the rest frame of particle or system $S$, and $m_X^{-1}\lambda(m_X; m_Y, m_Z)$ is the phase space volume for 2-body decay with mother mass $m_X$ and daughter masses $m_Y$ and $m_Z$. The form of $\lambda(m_X; m_Y, m_Z)$ is given in (16). Note that the phase space volume element is independent of the angular coordinates.

We must specify how these coordinates are to be mapped onto the output hypercube of the ANN. The ranges of the two invariant mass coordinates $m_{234}$ and $m_{34}$ are given by

$$m_{234} \in \left(m_2 + m_3 + m_4, \sqrt{s} - m_1\right), \quad m_{34} \in \left(m_3 + m_4, m_{234} - m_2\right), \tag{5}$$

where $s$ is the square of the total center of mass energy of the process. We then assign these ranges uniformly to two coordinates $x_1$ and $x_2$ on the unit hypercube. The differential $dm_{234}$ in the phase space (4) then becomes $dm_{234} = (m_{234}^{\max} - m_{234}^{\min})\, dx_1$, and similarly for $dm_{34}$ and $x_2$. Note that when $m_{234} = m_{234}^{\min}$, the range of $m_{34}$ shrinks to zero. Thus the phase space weight in terms of the hypercube coordinate $x_2$ is zero when $x_1 = 0$ (corresponding to the minimum value of $m_{234}$). However, this is handled well by our training algorithm thanks to the clipping procedure introduced in Section 2.

For the purposes of mapping the angular coordinates onto the output hypercube, we fix the orientation of particle 1 in an arbitrary direction. This fixing corresponds to two angles of overall rotation represented by $d\Omega^{(X)}$ in (4). Specific values may be chosen later if desired for generating simulated events[3]. Particle 2 makes some angle with respect to particle 1 in the (234) frame. It is natural to use the cosine of this angle as a coordinate because it appears in the measure on the sphere in this frame $d\Omega^{(234)} = d\cos\theta_{12}^{(234)}\, d\psi$. We fix the azimuthal rotation of particle 2 around particle 1 in an arbitrary direction, which corresponds to the last overall rotation angle $\psi$ that we left free. In the next stage of the decomposition, we must specify the orientation of particle 3 in the (34) frame. This can be done in terms of the cosine of the polar angle with respect to particle 2, $\cos\theta_{23}^{(34)}$, and an azimuth $\phi$ measured from the plane defined by the direction of particle 1 in this frame. This choice corresponds to the factor $d\Omega^{(23)}$ in (4). Having defined our angular coordinates in this way, the ranges $[-1, 1]$ of the cosines of $\theta_{12}^{(234)}$ and $\theta_{23}^{(34)}$ and $[0, 2\pi]$ of the azimuthal angle $\phi$ can be mapped uniformly onto the three remaining coordinates on the unit hypercube.

Although the phase space weight only depends on two invariant masses, important features in the matrix element such as resonances or collinear singularities may appear in terms of the invariant mass of any set of final particles. We therefore need a simple way to compute the invariant mass of any pair in terms of our phase space coordinates. These relations are given in Appendix A.

In Appendix B we present an alternative method of sampling the three angular coordinates that is symmetrical and in which all angles are defined in the same frame.

---

[3]Because we are considering only the decay of an on-shell Higgs, there is no intrinsic direction in the problem, and so the phase space measure with respect to $d\Omega^{(X)}$ is uniform. However, if the overall orientation of the event is correlated with some direction such as the accelerator beam, this could be included.

# 4 Results: Higgs Decay to Four Leptons

In this section, we present the results of an ANN-based simulation of the on-shell Higgs decay into 4 leptons $\mu^+, \mu^-, e^+, e^-$ with two intermediate $Z$ bosons. We label the $\mu^+$ as particle 1, $\mu^-$ as particle 2, $e^+$ as particle 3, and $e^-$ as particle 4. The differential decay width of this process, using our parametrization of the 4-body phase space, can be written as

$$d\Gamma = m_h^{-1} |\overline{\mathcal{M}}|^2 \, d\Pi_4 \,, \tag{6}$$

where $m_h$ is the Higgs mass, $|\overline{\mathcal{M}}|^2$ is the spin-summed invariant matrix element-squared, and $d\Pi_4$ is the phase space volume element given above. Assuming the leptons are massless, the tree-level matrix element of this process is given by [29]

$$
\begin{aligned}
|\overline{\mathcal{M}}|^2 &= |\mathcal{M}^{+-+-}|^2 + |\mathcal{M}^{+--+}|^2 + |\mathcal{M}^{-++-}|^2 + |\mathcal{M}^{-+-+}|^2 \\
\mathcal{M}^{+-+-} &= \frac{2e^3 g^+_{f_1 f_2} g^+_{f_3 f_4} \mu_W}{c_W^2 s_W} \frac{\langle k_1 k_3 \rangle^* \langle k_2 k_4 \rangle}{(m_{12}^2 - \mu_Z^2)(m_{34}^2 - \mu_Z^2)} \\
\mathcal{M}^{+--+} &= \frac{2e^3 g^+_{f_1 f_2} g^-_{f_3 f_4} \mu_W}{c_W^2 s_W} \frac{\langle k_1 k_4 \rangle^* \langle k_2 k_3 \rangle}{(m_{12}^2 - \mu_Z^2)(m_{34}^2 - \mu_Z^2)} \\
\mathcal{M}^{-++-} &= \frac{2e^3 g^-_{f_1 f_2} g^+_{f_3 f_4} \mu_W}{c_W^2 s_W} \frac{\langle k_2 k_3 \rangle^* \langle k_1 k_4 \rangle}{(m_{12}^2 - \mu_Z^2)(m_{34}^2 - \mu_Z^2)} \\
\mathcal{M}^{-+-+} &= \frac{2e^3 g^-_{f_1 f_2} g^-_{f_3 f_4} \mu_W}{c_W^2 s_W} \frac{\langle k_2 k_4 \rangle^* \langle k_1 k_3 \rangle}{(m_{12}^2 - \mu_Z^2)(m_{34}^2 - \mu_Z^2)} \,,
\end{aligned}
\tag{7}
$$

where $c_W$ and $s_W$ are the cosine and sine of the weak mixing angle, and $\langle k_i k_j \rangle$ is the spinor bracket with $|\langle k_i k_j \rangle| = m_{ij}$. The complex masses of the $Z$ and $W$ bosons $\mu_Z$ and $\mu_W$ are given by

$$\mu_V^2 = M_V^2 - i M_V \Gamma_V \,, \quad V = Z, W \,, \tag{8}$$

where $M_V$ are the physical masses and $\Gamma_V$ are the decay widths. The coupling constants $g^\pm_{ff}$ are given by

$$g^+_{ff} = -\frac{s_W}{c_W} Q_f \,, \quad g^-_{ff} = -\frac{s_W}{c_W} Q_f + \frac{I^3_{W,f}}{c_W s_W} \,, \tag{9}$$

where $Q_f$ is the electric charge of the fermion $f$ and $I^3_{W,f}$ is the third component of its weak isospin.

The ANN is constructed and trained as described in Sec. 2. A set of $10^7$ events is generated by the trained ANN, and the unweighting procedure is performed on this set. The unweighting efficiency is 26%, an improvement of about a factor of three compared to 8% efficiency for the same process acheived by `MadGraph`. Distributions shown in Figs. 5 and 6 are based on the set of $2.6 \times 10^6$ unweighted events. Invariant mass and angular distributions generated by the ANN, shown in Fig. 5, are in excellent agreement with `MadGraph` results. (We also checked that ANN distributions agree precisely with those generated by uniform sampling of phase space, an extremely inefficient but reliable Monte Carlo technique.) Furthermore, the two-dimensional density plots in Figs. 6 show that the ANN simulation reproduces the expected resonance structure. This includes both a resonance in the kinematic variable aligned with one of the target-space coordinates ($m_{34}$), and one in the variable not aligned with any of the target-space coordinates ($m_{12}$). The ability of the ANN to reproduce such non-aligned resonances is an important advantage of this approach, which may become increasingly important for simulating processes with more complex structure of resonances and singularities, for example at higher orders in perturbation theory.



Figure 5: Kinematic distributions of $h \to 4\ell$ decays in the five coordinates chosen to parametrize the final-state phase space (a-e) and in the invariant mass $m_{12}$, which is not aligned with any of the chosen coordinates but contains a resonance (f). The coordinates have been rescaled from the unit hypercube back to their original values. The output of the ANN-based simulation (after unweighting) is represented by orange solid histogram. For comparison, distributions generated by `MadGraph` are shown by black lines. Residuals are shown beneath each plot. The grey bands are statistical uncertainties from random sampling of the phase space and event unweighting.

Figure 6: Density plots of $2.6 \times 10^6$ points generated by the ANN (after unweighting). Plotted kinematic variables are aligned with coordinates on the target-space hypercube, with the exception of $m_{12}, m_{13}, m_{24}$.

## 5 Bijectivity of the ANN Map

The map $I \mapsto T$ defined by a fully-connected ANN is not automatically guaranteed to be bijective[4]. Lack of bijectivity can cause significant issues in the context of MC simulation:

- If the map is not surjective, there will be regions in phase space where no events are generated, regardless of the sample size.

- If the map is not injective, *i.e.* the map $T \mapsto I$ inverse to $\mathbf{y_w(x)}$ is multi-valued, a small

---

[4]A bijective fully-connected network can be constructed, see *e.g.* [30], but bijectivity requires the same number of nodes in all layers, which does not appear to be well-suited to the problem at hand.

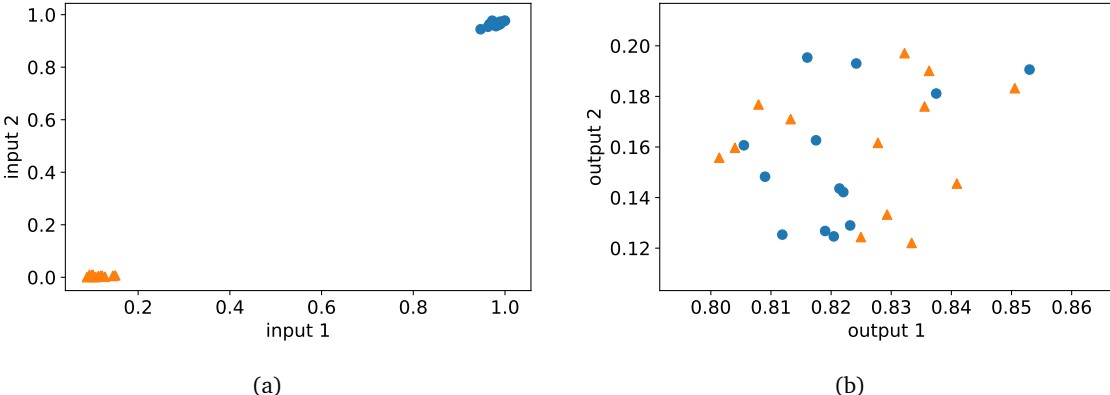

Figure 7: An example of a non-injective region in the "comparison sample" map generated by an ANN trained with a loss function that favors foldings, see Eq. (13). Points in two distinct regions in the input space (a) map onto the same region in the output space (b).

region in $T$ may be populated by points within two or more clusters in $I$, as illustrated for example in Fig. 7. In this case, the phase space density computed according to Eq. (1) at each $\mathbf{y}_i$ is incorrect, potentially invalidating both our training algorithm and unweighting procedure.

Fortunately, training with the KL distribution loss function tends to naturally prefer bijective (or at least approximately bijective) maps:

- **Surjectivity:** Note that the integral of the induced pdf over the target space is fixed:

$$\int_T d\mathbf{y}\, p_y(\mathbf{y}) = \int_T d\mathbf{y} \left| \frac{\partial x_j}{\partial y_i} \right| = V_I = 1. \tag{10}$$

  If the target pdf is normalized to integrate to one as well, a non-surjective map would necessarily result in mismatched normalization between induced and target pdfs in the region of phase space covered by the map. The minimum of the loss function, $L_{KL} = 0$, is reached when the two pdfs have the same normalization, *i.e.* for a surjective map. If the target pdf is *not* normalized, it can always be rewritten as $f(\mathbf{y}) = C f_N(\mathbf{y})$, where $f_N$ is a normalized pdf and $C$ is a constant. Using (1) and (10), it is easy to show that the minimum of the loss function in this case is given by $L_{KL} = -\log C$, and is reached when $p_y = f_N$ and the map is surjective.

- **Injectivity:** Since the map defined by the ANN is always continuous, lack of injectivity necessarily results in some phase space regions with very small Jacobians and thus very large induced probability density (see Fig. 8 for an illustration in 1 dimension). Since the target pdf is generic in these regions, such features are generally strongly penalized by the loss function. Training would therefore tend to "smooth out" the foldings, resulting in an injective map.

We rely on the above features to produce a bijective map through training, and test the trained ANN for bijectivity *a posteriori*. The rest of this section describes the tools used for this test, and the results showing that the ANN trained to simulate $h \to 4\ell$ decays is bijective to a good approximation.

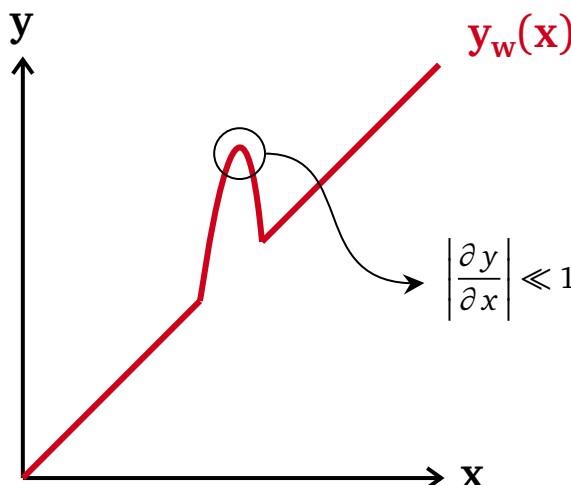

Figure 8: A one-dimensional illustration of why a non-injective region ("folding") in the map necessarily gives rise to a small Jacobian.

## 5.1 Surjectivity

While unweighting efficiency is a common measure of simulation quality, it cannot be used to test surjectivity. The unweighting efficiency is evaluated using only points present in the sample. If those points follow the shape of the target distribution perfectly in the portion of the phase space that is covered, unweighting efficiency would be equal to one, even if there are regions of phase space with non-zero target pdf that remain completely uncovered. To assess surjectivity of the map, we instead use the value of the pdf integral implied by the generated sample:

$$I_{\text{MC}} = \frac{1}{N} \sum_{\mathbf{y}_i} w_r(\mathbf{y}_i), \tag{11}$$

where the sum is over the $N$ points in the sample, and $w_r(\mathbf{y}_i) = f(\mathbf{y}_i)/p_y(\mathbf{y}_i)$ are the raw weights. The "true" value of the integral $I_{\text{true}}$ can be found for example by using uniform sampling of phase space, which is very inefficient as an MC generator but is guaranteed to cover the full phase space. Lack of surjectivity is indicated by $I_{\text{MC}} < I_{\text{true}}$. For the $h \to 4\ell$ simulation presented in Section 4, we find $I_{\text{MC}}/I_{\text{true}} = 0.993$, indicating that the trained ANN map is surjective to a good approximation. The remaining small regions not covered by the ANN map lie at the phase space boundaries, which accounts for the discrepancy between ANN and MadGraph in the very last bin of the $\cos\theta_{23}^{(34)}$ distribution in Fig. 5 (d).

## 5.2 Injectivity

Lack of injectivity means that disparate clusters in the input space get mapped into the same phase space region, see Fig. 7. We refer to this situation as "folding." To diagnose whether foldings are present in a trained ANN map, we divide the output space $T$ into a large number of small hypercubes. For each small hypercube, we calculate the covariance matrix of the coordinates of the points in the input space that map into it. Eigenvalues of the covariance matrix are then found, and the ratio $R$ of the largest to second-largest eigenvalue is used as an indicator of possible folding. For a bimodal distribution such as in Fig. 7, the separation of the two clusters defines a preferred direction which will be associated with a large eigenvalue. In this case the $R$-value is expected to be large, while for an injective map it is generally of order one. An exception may occur if an injective map happens to project an oblong (but singly connected) region in $I$ into the same hypercube, as shown for example in Fig. 9. Nevertheless,

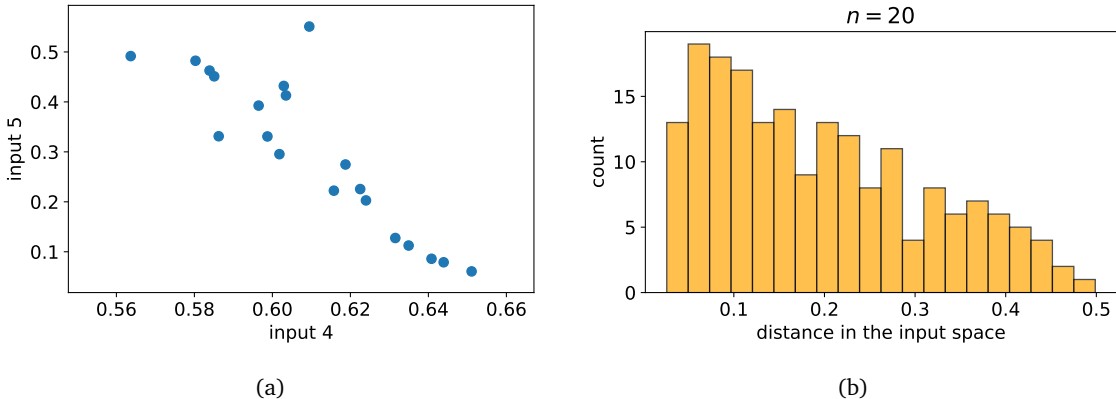

(a)                                          (b)

Figure 9: Input space points that map into the hypercube with the largest $R$-value in
the main sample, $R = 54$. (a) Oblong shape of the distribution is responsible for the
large $R$ value. (b) Pairwise distances among the input-space points show no evidence
for a bimodal distribution.

the $R$-value provides a useful diagnostic for foldings, in particular because it can be evaluated
efficiently with modest CPU time requirements. For hypercubes with $R$-values above some
threshold, we further evaluate the bimodal coefficient $b$ for the pairwise-distance distribution
of points in the input space, defined as [31]:

$$b = \frac{g^2 + 1}{k + \frac{3(n-1)^2}{(n-2)(n-3)}},\tag{12}$$

where $g$ is the sample skewness, $k$ is the sample excess kurtosis, and $n$ is the sample size.
A flat distribution would give $b = 5/9$, with larger $b$ indicating a bimodal distribution. (A
perfectly bimodal distribution with two delta-function clusters gives $b = 1$.) The bimodal
coefficient clearly distinguishes between "clustered" and "oblong" distributions[5], but is more
computationally expensive than the $R$-value. Finally, for a limited number of hypercubes where
folding is suspected based on both $R$ and $b$, histograms of pair-wise distances between points
in input space (such as for example in the right panel of Fig. 9) can be manually examined.

As a test of this diagnostic tool, it was applied to a sample of $10^7$ $h \to 4\ell$ events generated
by the trained ANN described above. A comparison sample was generated by using the same
setup, but training the ANN with a different loss function:

$$L(w) = L_{\text{KL}}(w) + L_{\text{int}}(w)$$
$$L_{\text{int}}(w) = -C \exp\left(-\frac{(I_{\text{MC}} - I_{\text{true}})^2}{2a^2}\right),\tag{13}$$

where $I_{\text{MC}}$ is the integral defined in Eq. (11), and $a$ and $C$ are constants.[6] The target space
was divided into $10^5$ hypercubes with side length of 0.1. To avoid noisy data from sparsely
populated phase space regions, hypercubes with fewer than 20 points were discarded. The
distribution of the $R$-values in the main and comparison samples is shown in Fig. 10. The
comparison sample contains hypercubes with very large values of $R$, indicating lack of in-
jectivity. Examination of input-space distributions in boxes with high $R$-values confirms that

---

[5]However, see Ref. [32] for a discussion of limitations of the bimodal coefficient.
[6]This alternative loss function was originally explored as a way to improve surjectivity by including the integral
explicitly in the training procedure, but was ultimately not used due to its adverse effect on injectivity.

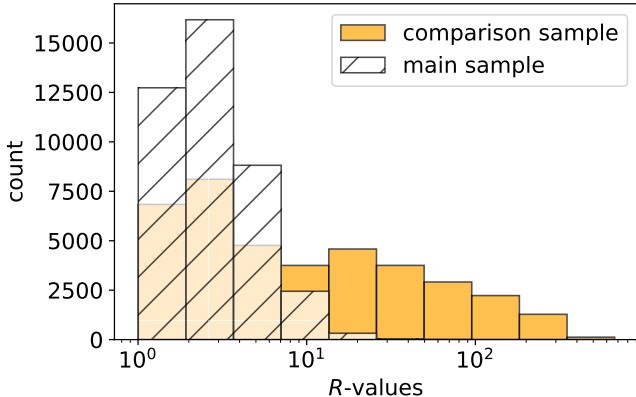

Figure 10: Distribution of $R$-values in the main sample (hatched) and the comparison sample (orange).

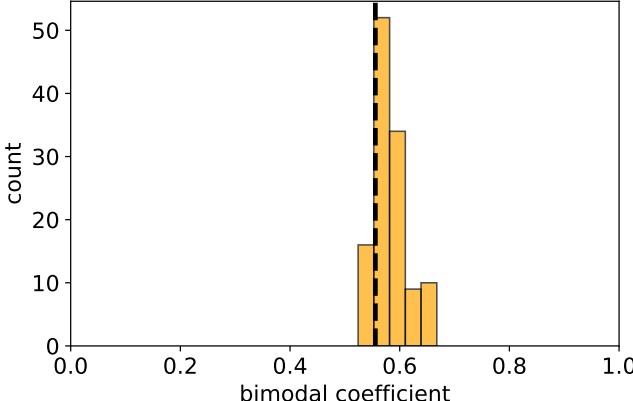

Figure 11: Distribution of the bimodal coefficient for hypercubes with the largest $R$-values (20 and above) in the main sample. No significant deviations from the flat-distribution value $b = 5/9$ (shown by a dashed line) are observed, indicating absence of foldings.

foldings are indeed present; for example, this is clearly visible in Fig. 7 for a cube with the largest $R$ value in the comparison sample, $R = 670$. In contrast, the main sample contains few hypercubes with large $R$. The maximum $R$-value in this sample is 54, and results from an oblong input-space distribution rather than folding, see Fig. 9. The values of $b$ for the 120 boxes with largest $R$-values in the main sample, shown in Fig. 11, cluster narrowly around the flat-distribution value, so there is no indication of foldings from this measure. As an additional test, we manually examined the pairwise input-space distances in boxes with largest $b$, and did not find any evidence of folding. Based on this data, we conclude that while the map used to generate the comparison sample is clearly not injective, the main sample shows no sign of deviations from injectivity.

While this conclusion is reassuring, foldings may of course still occur in the main sample at length scales shorter than 0.1. In general, foldings at smaller scales have progressively smaller effect on the quality of the simulation, while also being more computationally expensive to diagnose. We plan to address this issue more fully in future work.

# 6 Conclusion and Outlook

In this work, the ANN-based Monte Carlo generator proposed in [1] has been applied to simulate the Higgs decay to four charged leptons, a process of great interest for the LHC experiments. A convenient parametrization of the four-body phase space, mapping it into a unit hypercube which is a natural output space for ANN, was developed for this application. A numerical instability was encountered during the training process. This instability arises due to the structure of the four-body phase space. The training algorithm was supplemented with "gradient clipping," which allows to avoid the instability and achieve stable convergence of the training process. The trained ANN was then used to generate a large sample of $h \to 4\ell$ events. The ANN simulation was shown to achieve high unweighting efficiency of 24% (compared to 8% for "off-the-shel" MadGraph simulation), and the integrated decay width in this channel is accurate to within 0.7%. The ANN simulation reproduced $Z$-boson resonances in both lepton pairs, including the one whose invariant mass was not aligned with any of the chosen phase space coordinates. The ability of the ANN to reproduce such non-aligned features offers a potentially powerful advantage over existing grid-based algorithms.

The map defined by a fully-connected ANN used in our algorithm is not automatically bijective, which may cause issues with simulation validity. Nevertheless, we have argued that the training algorithm prefers bijective maps. We developed numerical tools to check bijectivity *a posteriori*, such as the $R$-values and bimodal coefficients to identify phase space regions where lack of injectivity ("foldings") may occur. Using these tools, we did not find any sign of significant non-bijectivity in the trained ANN used on our $h \to 4\ell$ simulation.

The results presented here add to the evidence for the promise of ANN-based MC generation as a viable alternative to traditional algorithms such as VEGAS. In principle, the current algorithm can be applied to simulate any parton-level process, with arbitrary number of particles in the final state. To explore this further, it would be interesting to automate the ANN setup and training and to interface it with the matrix element generator. Another interesting direction would be to apply the algorithm to simulations beyond the leading order. Compared to the tree level, loop corrections to matrix elements have a more complicated analytic structure which includes branch cuts as well as poles, and the inherent ability of the ANN generator to find and reproduce features not aligned with coordinate axes may provide an important advantage in this case.

**Funding information** This research is supported by the U.S. National Science Foundation through grant PHY-1719877. MDK acknowledges support by the Samsung Science & Technology Foundation under Project Number SSTF-BA1601-07, and a Korea University Grant.

# A Computing Kinematic Invariants

In this appendix, we show how kinematic invariants that may enter the differential cross section/decay width may be expressed in terms of the subset of invariants and angles which were used to parametrize phase space for the ANN. We will make use of the "kinematic bracket" defined as

$$[A, B, C] = \frac{m_A^4 + m_B^4 + m_C^4 - 2(m_A^2 m_B^2 + m_B^2 m_C^2 + m_A^2 m_C^2)}{4 m_B^2}, \tag{14}$$

which gives the squared magnitude of the 3-momentum of either of the two particles with mass $m_A$ and $m_C$ involved in a 2-body decay in the rest frame of the third particle with mass $m_B$:

$$|\mathbf{p}_A|^2 = |\mathbf{p}_C|^2 = [A, B, C], \quad \mathbf{p}_B = 0. \tag{15}$$

$B$ may be the mother, in which case $\mathbf{p}_A$ and $\mathbf{p}_C$ point in opposite directions, or it may be a daughter, in which case they point in the same direction. The bracket is obviously symmetric under the exchange $A \longleftrightarrow C$. The 2-body phase space volume $\Pi_2$ can also be expressed in terms of the bracket as

$$m_B \Pi_2 = \lambda(m_B; m_A, m_C) \equiv \frac{\sqrt{[A,B,C]}}{4\pi}. \tag{16}$$

Suppose we have chosen to decompose the final state as shown in the left panel of Fig. 4, as we did in defining the coordinates for the ANN. Our two invariant mass coordinates are $m_{34}$ and $m_{234}$. Invariant masses are of course the same in any frame, and so we may choose to compute them in the most convenient frame. For example, we can easily find $m_{23}$ and $m_{24}$ by working in the rest frame of the (34) system. Using Eq. (14), we obtain the momenta

$$|\mathbf{p}_3|^2 = |\mathbf{p}_4|^2 = [3,34,4], \qquad |\mathbf{p}_2|^2 = [2,34,234]. \tag{17}$$

We chose one of the three angles required in our parametrization to be the angle $\theta_{23}^{(34)}$ between particles 2 and 3 in this frame. Thus we have

$$m_{23}^2 = m_2^2 + m_3^2 + 2\left(E_2 E_3 - |\mathbf{p}_2||\mathbf{p}_3| \cos\theta_{23}^{(34)}\right) \tag{18}$$

$$m_{24}^2 = m_2^2 + m_4^2 + 2\left(E_2 E_4 + |\mathbf{p}_2||\mathbf{p}_4| \cos\theta_{23}^{(34)}\right) \tag{19}$$

making use of the fact that particles 3 and 4 are back-to-back in this frame. Similarly, we can move to the (234) frame and find

$$|\mathbf{p}_2|^2 = [2,234,34], \qquad |\mathbf{p}_1|^2 = [1,234,X]. \tag{20}$$

Using the angular coordinate $\theta_{12}^{(234)}$, the invariant mass $m_{12}$ can be immediately computed.

It only remains to compute $m_{13}$ and $m_{14}$, but these are complicated in our current decomposition because the two particles in these invariants are separated in the decay chain by particle 2. Of course this is just an artifact of our choice of decomposition and we could just as well use the alternative illustrated in the right panel of Fig. 4. The last two invariant masses then follow in analogy with Eq. (18) and Eq. (19) in terms of the angle $\theta_{13}^{(34)}$. However, this was not one of the coordinates we originally selected. It can be found using the spherical law of cosines in terms of our last angular coordinate $\phi$ if we also know the angle between particles 1 and 2 in the (34) frame. This can be extracted from $m_{12}$ as

$$\cos\theta_{12}^{(34)} = -(|\mathbf{p}_1||\mathbf{p}_2|)^{-1}\left(\frac{m_{12}^2 - m_1^2 - m_2^2}{2} - E_1 E_2\right). \tag{21}$$

In computing $m_{13}$ and $m_{14}$, we also need to know the value of $m_{134}$ which is again not one of our original coordinates. However, by energy-momentum conservation, we know

$$\begin{aligned} m_{134}^2 &= \left(p_1^\mu + p_3^\mu + p_4^\mu\right)^2 = \left(p_X^\mu - p_2^\mu\right)^2 \\ &= m_X^2 + m_2^2 - 2\left(E_X E_2 - |\mathbf{p}_X||\mathbf{p}_2| \cos\theta_{12}^{(234)}\right). \end{aligned} \tag{22}$$

In the last step, we choose to work in the (234) frame where $|\mathbf{p}_1| = |\mathbf{p}_X|$ and we can use Eq. (20).

# B  Symmetric Phase Space Sampling

In Sec. 3, we described a method of sampling the angular coordinates in terms of the natural flat coordinates on the sphere at each stage of the decay. In the first stage, we sampled according to a flat distribution in $\cos\theta_{12}^{(234)}$. In the second stage, we sample according to a flat distribution in $\cos\theta_{23}^{(34)}$ and $\phi$. However, for computing kinematic invariants, which are likely to be important for the evaluation of the matrix element, it is necessary to have $\cos\theta_{13}^{(34)}$ as well. This can be computed from given values of $\cos\theta_{23}^{(34)}$ and $\phi$, but in this appendix we describe a way to sample phase space directly in terms of $\cos\theta_{13}^{(34)}$ and $\cos\theta_{23}^{(34)}$. In the following we will always work in the (34) frame, and drop the superscript. We also convert $\cos\theta_{12}^{(234)}$ into the (34) frame using Eq. (21). We indicate these three angular coordinates by $c_{12}$, $c_{23}$, and $c_{13}$.

It is straightforward to show that the measure on the sphere in the (34) frame can be written as

$$\mathrm{d}c_{23}\,\mathrm{d}\phi = \frac{\mathrm{d}c_{23}\,\mathrm{d}c_{13}}{\sqrt{1-c_{12}^2-c_{23}^2-c_{13}^2+2c_{12}c_{23}c_{13}}} = \mathrm{d}c_{23}\,\mathrm{d}c_{13}\,D^{-1/2}, \qquad (23)$$

where

$$D = \begin{vmatrix} 1 & c_{12} & c_{13} \\ c_{12} & 1 & c_{23} \\ c_{13} & c_{23} & 1 \end{vmatrix}. \qquad (24)$$

This formulation has the advantage that it is symmetric among the three angular coordinates. However, not all choices of $(c_{12}, c_{23}, c_{13}) \in [-1, 1]^3$ are physically possible. The condition for physical values of the coordinates is equivalent to requiring that the phase space weight is real, that is, $D > 0$. One may therefore sample these three angular coordinates in the 3-dimensional hypercube, discarding any points for which $D < 0$, and including the factor $D^{-1/2}$ in the target function.

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
