# Peer review of "Improved Neural Network Monte Carlo Simulation"

_SciPost Physics, doi:SciPost Phys. 10, 023 (2021)_

## Round 1 · Referee Report · Tilman Plehn (Referee 1) · 2020-11-6

Strengths

1- the paper is asking a very interesting and timely question; 2-it is very clearly written (much better than its earlier counterpart); 3- the discussion and conclusions are free of the usual unreasonable claims in the field....

Weaknesses

1- the study only looks at a relatively simple process, which is fine, though; 2- the obvious question is not answered, namely what's the advantage over invertible flows? 3- what happens with a variable number of final-state particles?

Report

It's a very nice paper, looking at a very relevant problem for the HL-LHC. Clearly, using ML inside the established MC generators is something we need to do and benchmark. The paper also nicely states that one of the lead problems is the unweighting probability. So it's all relevant and fine and publishable, with a few suggested changes to the text. The only big question the reader is left alone with is why this ANN should be used instead of the flow networks used by Sherpa. Is there any hope that it would solve the problems encountered in the two Sherpa studies? A good answer to this question would, to me, make the difference between just publishing the paper and publishing it as one of the top SciPost papers...

Requested changes

Just a list of changes in the current text, plus one important question: 1- in the introduction, it would be fair to mention mutil-channeling; 2- on p.4 the statement about the generator is a little unclear. It's not really a generator because the probabilistic aspect still needs unfolding, no? 3- what is the impact of the number of PS points per batch, mentioned on p.5? 4- is the described gradient clipping the standard/best solution? 5- why did you choose the phase space parametrization you are using? 6- related to this, are there PS boundaries not at 0 or 1, for instance in Fig.5? 7- how do you compute the unweighting efficiency and the comparison with Madgraph? 8- please add panels with ratios to Fig.5; 9- at the end of the conclusions, how would this tool help with higher orders? *- the elephant in the room: where is your method better than the invertible nets?

  • validity: high
  • significance: good
  • originality: good
  • clarity: top
  • formatting: perfect
  • grammar: perfect

Author:  Matthew Klimek  on 2020-11-24  [id 1058]

(in reply to Report 1 by Tilman Plehn on 2020-11-06)
Category:
answer to question

Thank you for your comments. We will respond here -- referenced changes to the draft will appear in a resubmitted version, which will be uploaded after we receive the remaining referee reports. In the meantime, any further comments or questions are welcome.

1-- We’ve added a mention of this in Introduction.

2-- We were unsure of what you are asking in this comment. We think you are referring to the fact that the ANN generates weighted events, although these can be easily unweighted to generate a sample of events. This process is described in Figure 2 and Section 2. Please let us know if we have misunderstood the question.

3-- We have added additional discussion on this in the relevant paragraph in Section 2.

4-- Clipping is a standard method for dealing with gradients that can become anomalously large, see for example Chapter 10.11 of the MIT Deep Learning text. It is a bit difficult to say precisely if it is the “best” but it is a standard and straightforward way to deal with the sort of issue we were seeing, and we find that it is very effective.

5-- There are two basic requirements for our phase space parametrization:
— It must be easily generalizable to any number of particles.
— The coordinates should easily map onto a hypercube.
Our choice satisfies both, and is in fact a variation on a standard parametrization that has been in use for decades (see, for example, https://cds.cern.ch/record/275743). This motivation is summarized in the first paragraph of Section 3.

6-- By construction there are no boundaries elsewhere. However in the plots for the angles (c, d, and e), we have rescaled the axis so that it corresponds to the usual range of cosine (-1,1). We amended the caption to make this clear.

7-- An explanation has been added to Section 2.

8-- Done.

9-- We have added an additional comment about this in the conclusion.

Re: flow networks--
In our opinion, both our technique and that used in the Sherpa studies are promising and deserve continued investigation. However, it is true that in both Sherpa studies, performance dropped sharply for 4 particles. In our study, we are able to handle 4 particles without a large drop in performance. While it would take more detailed studies to confirm definitively why this is the case, we can make one conjecture. That is, the normalizing flow / coupling layer -based studies ultimately make use of discrete maps — the target space is divided into a finite number of intervals and a fixed order interpolation is used within each interval. It has this feature in common with VEGAS in fact. What the normalizing flow studies add is that this mapping is done several times on different subsets of the coordinates, and at each step, the mapping is controlled by an ANN. It is thus much more flexible than VEGAS. Nevertheless, the ability to represent the target distribution must be limited by the finite number of intervals that are used, and this may be exacerbated as one goes to larger numbers of dimensions. In contrast, our method is fully continuous, and in that sense would not suffer from the same kind of limitations as the normalizing flows.
We have added a discussion of this to the Introduction.

---

## Round 1 · Referee Report · Stefan Höche (Referee 2) · 2020-11-18

Strengths

1 - The paper provides a thorough discussion of the techniques used to simulate four-particle decay processes and how to improve them with the help of artificial Neural Networks (ANNs)
2 - It includes a detailed analysis of possible instabilities in the training procedure that arise from jumps in the gradient norm which are due to coordinate singularities. The authors propose a simple clipping technique to remove these instabilities.
3 - A detailed analysis of the bijectivity of the ANN map is performed, finding approximate bijectivity, with possible deviations at small input space distances.

Weaknesses

1 - The phase space parametrization employed in the calculation is not optimal for the problem at hand.

Report

The authors present a detailed discussion of the four-particle decay phase space and how to improve event simulation for a four-particle decay with artificial neural networks (ANNs). Their new technique is applied to the highly relevant Standard Model off-shell decay h->4l and the event generation efficiency is compared to an off-the-shelf simulation with an automated event generator. A factor 3 improvement is found. The paper is sound and I recommend publication after a minor revision (see requested changes).

Requested changes

In order to put the results into perspective, it would be useful to also include density plots showing m_{12} vs m_{13} or m_{12} vs m_{14}. In standard event generators, the initial points are sampled from an off-shell decay squence X->{12->{1,2},3,4}, thus reflecting the dominant propagator structure of the amplitudes in Eq.(6). However, the spin correlations encoded in the numerator terms of Eq.(6) are usually not accounted for, and it would be instructive to see if the large improvement in event generation efficiency arises from the improved sampling of these terms alone.

  • validity: high
  • significance: good
  • originality: high
  • clarity: high
  • formatting: excellent
  • grammar: excellent

Author:  Matthew Klimek  on 2021-01-08  [id 1128]

(in reply to Report 2 by Stefan Höche on 2020-11-18)
Category:
remark

We thank the referee for a positive report on our paper. Following the referee’s suggestion, we have added density plots in pairs of invariant masses, m_{12} vs. m_{13} as well as m_{13} vs. m_{24}. See panels (e) and (f) in Fig. 6, in the version that will be resubmitted shortly. We think that it is highly likely that sampling according to the full matrix element is a major factor in improving the efficiency, as suggested by the referee. The best way to nail this down would be to compare the distributions produced by the NN and a traditional generator before unweighting, but unfortunately the MadGraph simulation used for comparisons in this paper does not provide access to the event sample at that level. We will try to address this point more fully in future work.

---

## Round 1 · Referee Report · Anonymous (Referee 3) · 2020-12-7

Strengths

The paper addresses important aspects left open in a previous study by a subset of the authors, namely the lack of bijectivity of the proposed ANN sampler. This it dealt with at sufficient detail in the present paper. The authors apply their novel sampler to the phenomenologically relevant decay of the SM Higgs to four leptons. The paper is clearly written and in fact a nice read.

Weaknesses

The discussion of the reference simulations done with MadGraph lack information about what technique is actually used by MadGraph to map the h->4l decay.

Report

The paper deserves publication in SciPost. It is clearly written and discusses the main criticisms raised against non-bijective NN sampling techniques. The application chosen is challenging and phenomenologically relevant.

Requested changes

Apart from the requests posted by the other commenters I have nothing specific to add. In particular I would be keen to see the ratio plots of Fig. 5, allowing one to judge if underpopulation of phase space is indeed marginal even in the tail of the distributions.

---

## Round 2 · Author Response

This version has been resubmitted with minor additions based on referees' comments. No results or conclusions have changed.

---

## Round 2 · List of Changes

The following additions have been made: - Mention of multi-channeling in first paragraph - Two new paragraphs at end of Introduction to discuss and contrast with other recent approaches in the literature - New paragraph containing Eq. 3 with more detailed explanation of unweighting and efficiency calculation - Expanded paragraph after Eq. 3 discussing choice of batch size for training - Residual plots in Fig. 5 - Additional slices of phase space in Fig. 6 - Additional sentence at end of Conclusions to discuss possible application to NLO simulation

---

## Editorial Decision

published